# Reduction of misdiagnosis in urinary tract infections during pregnancy: The role of adjusted urine flow cytometry parameters

**Zhaojie Liu[ID], Dan Liu, Guangming Su, Wei Yang[ID]***

Department of Laboratory Diagnostics, First Affiliated Hospital of Harbin Medical University, Harbin, Heilongjiang, China

* yangwei6295@163.com

**Data Availability Statement:** All relevant data are within the manuscript and its Supporting Information files.

## Abstract

### Introduction

Urinary tract infections (UTIs) pose a significant health concern, particularly among pregnant women, for whom accurate diagnosis is essential. However, the use of Urine flow cytometry (UF) for detecting UTIs in this demographic often results in misdiagnosis. The objective of this study was to explore the reasons behind these diagnostic errors and to develop a strategy to minimize the rate of UTI misdiagnosis in pregnant women.

### Material and methods

The study enrolled 1,200 women aged 18 to 40 years, categorized into pregnant and non-pregnant groups. UTIs were diagnosed using urine bacterial culture, microscopic examination, and UF, followed by statistical analysis to identify any discrepancies in diagnosis between the groups. Following the calibration of UF analyzer's parameters, the most effective CR(WBC)-CW-FSC-P Gain setting for diagnosing UTIs in pregnant women through UF was ascertained by applying the Youden index.

### Results

The clinical diagnosis rate of UTIs was significantly higher in pregnant women (40.91%) compared to non-pregnant women (20.26%). However, urine microscopy and bacterial culture showed no significant difference in the rates of UTIs between the two groups, suggesting a potential for misdiagnosis. The false-positive rate for WBCs detected by UF was 30.43%, and adjusting the CR(WBC)-CW-FSC-P Gain value of UF reduced the false-positive rate to 9.45%.

### Conclusion

The incidence of UTIs in pregnant women may be overestimated because of the limitations inherent to UF. Adjusting the parameters of the UF analyzer, particularly the CR(WBC)-CW-FSC-P Gain value, can significantly reduce the rate of UTI misdiagnosis in pregnant women.

**Funding:** The author(s) received no specific funding for this work.

**Competing interests:** The authors have declared that no competing interests exist

## Introduction

Urinary tract infections (UTIs) are a prevalent class of infections in clinical settings, posing a substantial health risk, particularly for pregnant women. These infections during pregnancy are correlated with preterm birth [1, 2]. Recognizing the profound impact of UTIs on both individual health and societal well-being, precision in the diagnosis of UTIs becomes indispensable [3]. However, the physiological changes that accompany pregnancy often complicate the accurate detection of UTIs, necessitating a more nuanced diagnostic approach [4, 5].

Leukocytes counts is a critical diagnostic criterion for UTIs [6, 7]. Urine flow cytometry (UF), recognized for its heightened sensitivity, has been extensively used for the detection of urinary leukocytes [8]. Obtaining a clean-catch midstream urine sample is essential for the accurate application of UF in detecting urinary leukocytes. However, urine samples from pregnant women often contain contaminants from the vaginal or anal regions, which can lead to false-positive or false-negative urinalysis results, and consequently, misdiagnoses [9]. Our previous study revealed that UF had a high false-positive rate when detecting urinary leukocytes in pregnant women [10]. The aim of this study was to elucidate the phenomenon that the high false positive rate of urinary leukocytes in pregnant women may lead to misdiagnoses of UTIs, with the ultimate goal of identifying a method to reduce the misdiagnosis rate in pregnant women with UTIs.

## Material and methods

The present study was approved by the Medical Ethics Committee of the First Affiliated Hospital of Harbin Medical University (No.2022103) and was conducted in accordance with the Declaration of Helsinki. All methods were carried out in accordance with the relevant guidelines and regulations. A written informed consent was taken from every participant of the study.

### Study population

A total of 1,200 women, aged between 18 and 40 years, who visited the obstetrics and gynecology outpatient department from June 1, 2023, to January 1, 2024, participated in this study. All participants included in this study underwent urine human chorionic gonadotropin (HCG) testing and abdominal ultrasound examination. Participants were categorized as pregnant if the urine HCG test was positive, or if the abdominal ultrasound revealed the presence of a gestational sac and fetal heartbeat within the uterine cavity. Conversely, if the urine HCG test was negative and the abdominal ultrasound did not detect a gestational sac or fetal heartbeat, participants were classified as non-pregnant. According to this criterion, we anticipate that there will be 600 participants in the pregnant group and an equal number in the non-pregnant group. Detailed selection and screening process is shown in Fig 1 Exclusion criteria included the use of indwelling catheters, immunosuppressive agents, and antimicrobial drugs, as well as an inability to express symptoms. For a diagnosis of UTI, patients should present with recent-onset urinary tract irritative symptoms, which include increased frequency of urination, a sudden compelling need to urinate, or painful urination. In addition to these clinical manifestations, the diagnostic criteria were satisfied by the fulfillment of at least one of the following objective parameters:

1. The detection of five or more leukocytes per high-power field ($\geq$5/HPF) within the urine sediment.

2. A positive urine culture that confirms the presence of a single uropathogen with a concentration of $10^4$ or more colony-forming units per milliliter ($\geq 10^4$ CFU/mL).

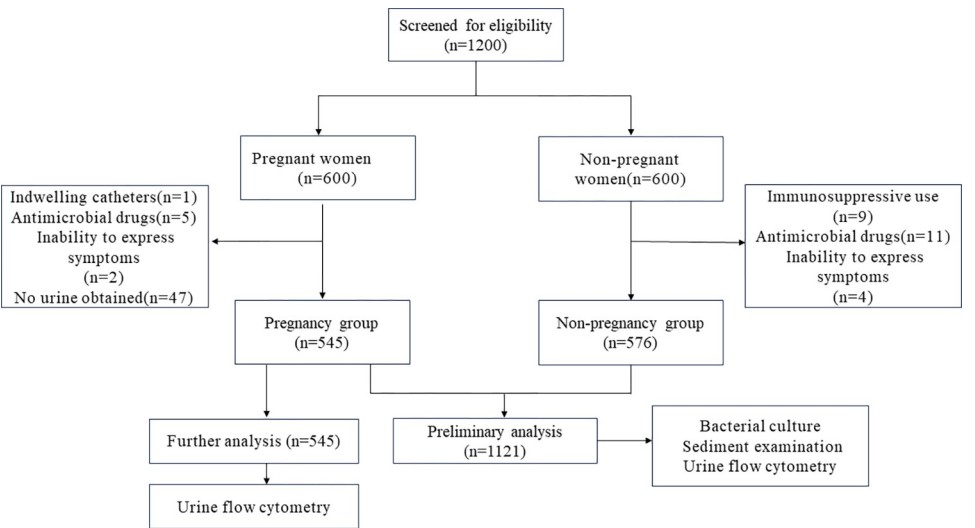

**Fig 1. Overview of screening and selection process.**

## Urinalysis

A 20mL midstream urine sample was randomly collected, mixed evenly, and then divided equally into two tubes, with each tube containing 10 mL. One tube was used for the urine bacterial culture and urinary sediment examination, and the other tube was used for UF.

## Microbiological examination

10 microliters of non-centrifuged urine were placed on blood agar plates, and the results were observed after 24 hours. A culture result was deemed positive in case of growth of $10^4$ CFU/mL or more and defined as a monoculture if 90% or more of the cultured colonies were of 1 microorganism. The primary pathogens isolated from the urinary tract are predominantly Escherichia coli, Enterococcus species, Klebsiella spp., and Proteus mirabilis. The growth of more than three bacterial species on the agar plate was indicative of contamination.

## Microscopic examination

The remaining urine specimens were centrifuged in a horizontal centrifuge at 400g for 5 minutes. The supernatant was discarded, and the remaining sediment was mixed. Next, 0.2 mL of the urine sediment was collected using a pipette and placed on a glass slide, which was then covered with an 18 × 18 mm coverslip. Under a light microscope, the number of white blood cells (WBCs) and epithelial cells (ECs) were counted at high magnification for 10 consecutive visual fields, and the average number per visual field was calculated. Each sample was counted by two veteran morphologists under double-blind conditions. For the microscopic method, the following diagnostic cut-off values were used for WBCs and ECs: negative, ≤5/HPF; positive, >5/HPF.

## Urine flow cytometry

UF was conducted utilizing the Sysmex UF-5000 analyzer (Sysmex Corporation, Kobe, Japan). The Sysmex UF-5000's initial settings for the compensated forward scatter (CW-FSC-P), side scatter height (CW-SSH-P), fluorescence height (CW-FLH-P), and differential side scatter (CW-DSS-P) parameters within the coefficient ratio (CR) for the WBCs were established at

0.88, 1551, 2428, and 2600, respectively. Subsequently, an assessment of the UF-5000's diagnostic performance was executed with a range of CW-FSC-P Gain values, specifically at 0.78, 0.98, 1.08, and 1.18, under the condition that all other parameters remained unchanged. The Gain value was adjusted mainly according to the change of Youden index. In the UF5000 analyzer, the initial setting for the gain value was established at 0.88. Upon adjusting the gain value to 0.78, we observed an increase in the false positive rate and a concurrent decrease in the Youden index. This finding indicated that reducing the gain value did not effectively lower the false positive rate of the UF5000. Subsequently, we incrementally increased the gain value to 0.98, 1.08, and 1.18. The selection of urine samples for analysis was predicated on predefined thresholds, where samples with a WBC count $\geq 25/\mu L$ and an epithelial cell (EC) count of 31 or higher per microliter ($\geq 31/\mu L$) were categorized as positive. Samples that did not meet these thresholds were designated as negative.

## Statistical analysis

Statistical analysis was conducted using SPSS version 26 (SPSS, Inc., Chicago, USA). The data were presented as the number of cases (n), percentages, and means with standard deviations. Microscopic examination was used as the gold standard for diagnosis in this study. When a sample was positive by other tests but negative by microscopy, it was called a false positive. Conversely, when a sample was negative by other tests but positive by microscopy, this was called a false negative. The positive rates of UTIs between pregnant and non-pregnant women were compared using chi-square tests. WBC results from false positive samples and ECs results from false negative samples, as determined by the UF-5000 and the microscopic method, were compared utilizing Pearson's correlation coefficient. The receiver operating characteristic (ROC) curves were plotted using GraphPad Prism version 9.3.1 (GraphPad Software, San Diego, California, USA) to assess the diagnostic performance of the index tests with different Gain values for CW-FSC-P. The area under the curve (AUC) was calculated to determine the discriminatory power of these tests. The optimal cutoff values, balancing sensitivity and specificity, were identified using the Youden index. For all analyses, $P < 0.05$ was defined as statistically significant.

## Results

From an initial screening of 1,200 participants, 1,121 individuals met the inclusion criteria and were categorized into two groups: the pregnancy group with 545 participants and the non-pregnancy group with 576 participants (Fig 1). A comprehensive summary of the baseline characteristics for the study groups is presented in Table 1. The groups were found to be demographically similar with respect to age, representation of certain ethnic groups (Han, Korean, Mongolian), and prevalence of hypertension. However, significant disparities were observed between the two groups in terms of other ethnic backgrounds (Manchu, Hui, Xibe, and Daur), smoking habits, alcohol consumption, and incidence of diabetes. Notably, the pregnancy group reported no cases of alcohol consumption, in contrast to 45 instances in the non-pregnancy group.

## The prevalence of UTIs among pregnant women is observed to be higher than that among non-pregnant women

Clinical diagnosis within the pregnant cohort revealed a significantly higher rate of UTIs (40.91%), when juxtaposed with a 20.26% rate observed in the non-pregnant cohort ($P < 0.001$), as detailed in Table 2. UF demonstrated a positive rate for WBCs of 41.83% in the pregnant group, which was markedly higher than the 20.16% rate in the non-pregnant group,

**Table 1. Baseline characteristics.**

|  | Pregnancy group (n = 545) | Non-pregnancy group (n = 576) |
|---|---|---|
| **Age (Years)** | 31.38 (4.29) | 33.09(3.98) |
| **Gestational age (Weeks)** | 29.02 (8.20) | – |
| **Race(n)** |  |  |
| _Han_ | 514 | 532 |
| _Korean_ | 11 | 14 |
| _Manchu_ | 8 | 13 |
| _Mongolian_ | 5 | 6 |
| _Hui_ | 5 | 11 |
| _Xibo_ | 1 | 0 |
| _Daur_ | 1 | 0 |
| **Smoking habit (n)** | 1 | 11 |
| **Alcohol consumption (n)** | 0 | 45 |
| **Diabetes mellitus(n)** | 92 | 51 |
| **Hypertension(n)** | 54 | 59 |

with this discrepancy being statistically significant (P < 0.001). These findings are congruent with clinical diagnoses and underscore the diagnostic value of WBC presence in urine for UTIs. However, urine microscopy, which is traditionally considered the gold standard for urine sediment analysis, showed a positive rate for WBCs of 22.75% in the pregnant group and 19.51% in the non-pregnant group, indicating no significant difference between the two cohorts (P > 0.05) (Table 2). These findings suggest that the increased rate of UTI diagnosis in pregnant women could be attributable to diagnostic misattribution, potentially associated with the outcomes of the UF examination. This hypothesis is supported by the data presented in Table 3, which shows that UF produced 126 false positives for WBC detection in the pregnant group, corresponding to a false positive rate of 30.43%. This indicates that the elevated incidence rate of UTIs in pregnant women may be a result of false positives during WBC detection by UF, consequently leading to clinical misdiagnosis. The diagnostic accuracy of UF in the context of UTIs within the pregnant population, as illustrated in Fig 2A, was inferior to that in the non-pregnant population. Furthermore, Fig 2B demonstrates that UF is less effective than urine sediment microscopy examination, highlighting the suboptimal diagnostic efficacy of UF in this specific demographic.

## Misdiagnosis potentially stems from urine sediment analyzer misidentifying epithelial cells as white blood cells

UF has been found to identify 130 instances of false negatives in the detection of ECs within urine samples of pregnant women, resulting in a false negative rate of 33.59%. This rate is notably similar to the false positive rate for WBCs, which is 30.43%, as shown in Table 3. This

**Table 2. Diagnosis of urinary tract infections in pregnant and non-pregnant women using different methods.**

|  | Positive rate (%) | | | Negative rate (%) | | |
|---|---|---|---|---|---|---|
|  | Pregnancy group | Non-pregnancy group | P-value | Pregnancy group | Non-pregnancy group | P-value |
| **Clinical Diagnosis** | 40.91 | 20.26 | <0.001 | 59.09 | 79.74 | <0.001 |
| **urine flowcytometry** | 41.83 | 20.16 | <0.001 | 58.17 | 79.84 | <0.001 |
| **microscope** | 22.75 | 19.51 | >0.05 | 77.25 | 80.49 | >0.05 |
| **bacteriological culture** | 17.92 | 16.09 | >0.05 | 82.08 | 83.91 | >0.05 |

**Table 3. The performance of urine flowcytometry (UF) and microscopic method in screening urinary WBC and EC of pregnant women.**

| | Positive for both UF and microscopic (n) | Negative for both UF and microscopic (n) | UF positive/ microscopic negative (n) | UF negative/ microscopic positive (n) | Positive predictive value (%) | False positive* rate (%) | Negative predictive value (%) | False negative# rate (%) |
|---|---|---|---|---|---|---|---|---|
| WBC | 102 | 288 | 126 | 29 | 44.73 | 30.43 | 90.85 | 22.13 |
| EC | 257 | 139 | 19 | 130 | 93.11 | 14.84 | 51.67 | 33.59 |

*False positive: Positive by urine flowcytometry but negative by microscopy.

#False negative: Negative by urine flowcytometry but positive by microscopy.

congruence suggests that there may be a correlation between the false positive rates for WBCs and false negative rates for ECs in UF detection. To empirically examine this hypothesis, a linear correlation analysis was conducted. This study assessed the relationship between the number of WBCs detected by UF and the corresponding count of ECs observed through microscopy in a sample of 126 cases that resulted in false positive WBC readings (Fig 3A). The analysis produced a correlation coefficient of r = 0.63 (P < 0.0001), indicating a significant positive correlation. In parallel, another linear correlation analysis was executed to evaluate the relationship between the number of ECs observed microscopically and the number of WBCs identified by UF in 130 cases that were false negatives for ECs (Fig 3B). This analysis yielded a correlation coefficient of r = 0.53 (P < 0.0001), also denoting a significant positive correlation. The results of these analyses collectively suggest that the misattribution of ECs as WBCs by UF may be a contributing factor to the elevated false positive rate for WBCs in urinalysis of pregnant women.

## Adjusting the CR(WBC)-CW-FSC-P Gain value of UF reduces the misdiagnosis rate of UTIs in pregnant women

The elevated rate of false positives may correlate with the utilization of suboptimal settings for the instrument's parameters. A potential resolution to the inflated false positive rate, which arises from the misclassification of epithelial cells as WBCs by UF, involves the recalibration of the detection parameters. Post-adjustment of the CR(WBC)-CW-FSC-P Gain value on the UF instrument, the consequent enhancement in the analyzer's efficacy for the detection of urinary WBCs in pregnant women is depicted in Fig 4.

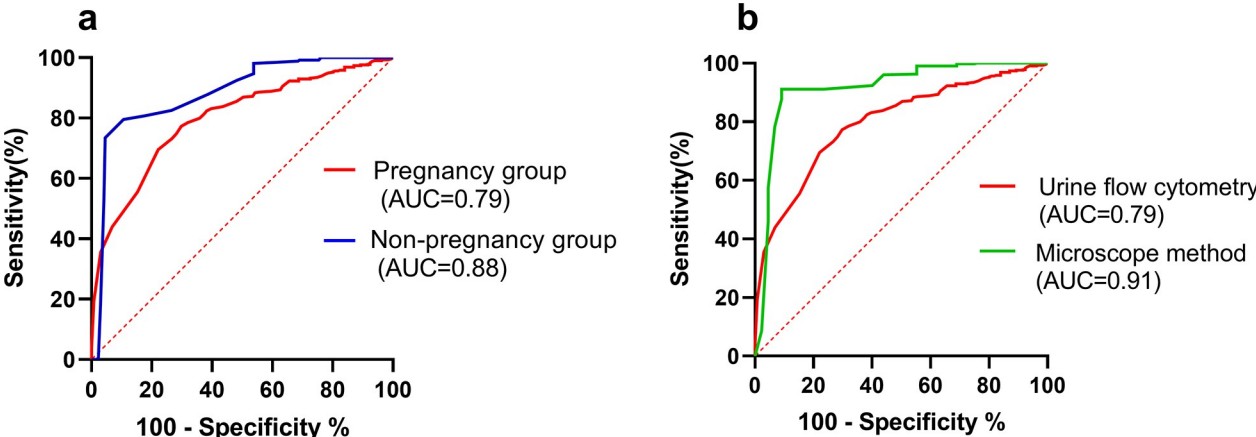

**Fig 2. Receiver operating characteristic (ROC) curves.** ROC curves for urine flow cytometry in diagnosing Urinary Tract Infections (UTIs) in both pregnancy and non-pregnancy groups (a). ROC curves for urine flow cytometry and the microscope method in diagnosing UTIs in pregnancy (b).

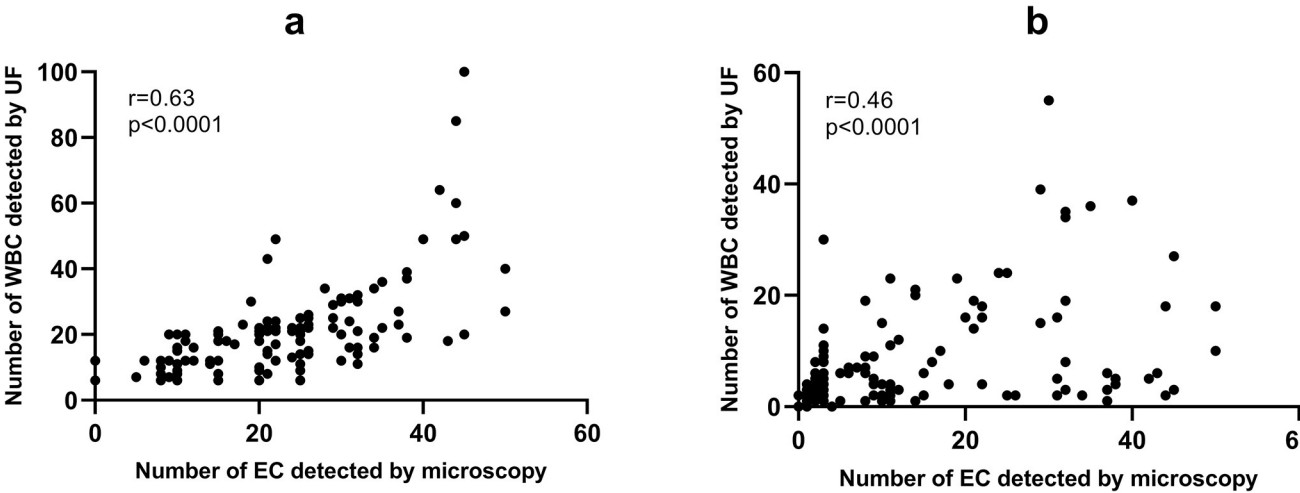

**Fig 3. Correlation analysis.** Correlations between the number of WBCs detected by UF and the corresponding number of epithelial cells observed microscopically in 126 cases of false positive WBCs samples (a). Correlations between the number of epithelial cells observed microscopically and the number of WBCs detected by UF in 130 cases of false negative epithelial cells samples (b).

Our analysis revealed that the AUC, a metric for diagnostic efficacy, escalates with an increase in the Gain value, attaining an apex of 0.91 at a Gain value of 1.08, beyond which there is a subsequent decline. Table 4 provides a detailed breakdown of the false positive rate for WBCs in relation to various CR(WBC)-CW-FSC-P Gain values. Notably, upon setting the Gain value to 1.08, the false positive rate for WBCs was substantially reduced to 9.45%, concomitant with an optimal Youden's index of 65.33%.

These findings indicate that by precisely adjusting the CR(WBC)-CW-FSC-P Gain value on the UF instrument, it is possible to significantly reduce the incidence of false positives for WBCs, thereby decreasing the likelihood of misdiagnosing UTIs in pregnant women. Specifically, when the CR(WBC)-CW-FSC-P Gain value was fine-tuned to 1.08, the diagnostic accuracy of UF for UTIs in this demographic was maximized.

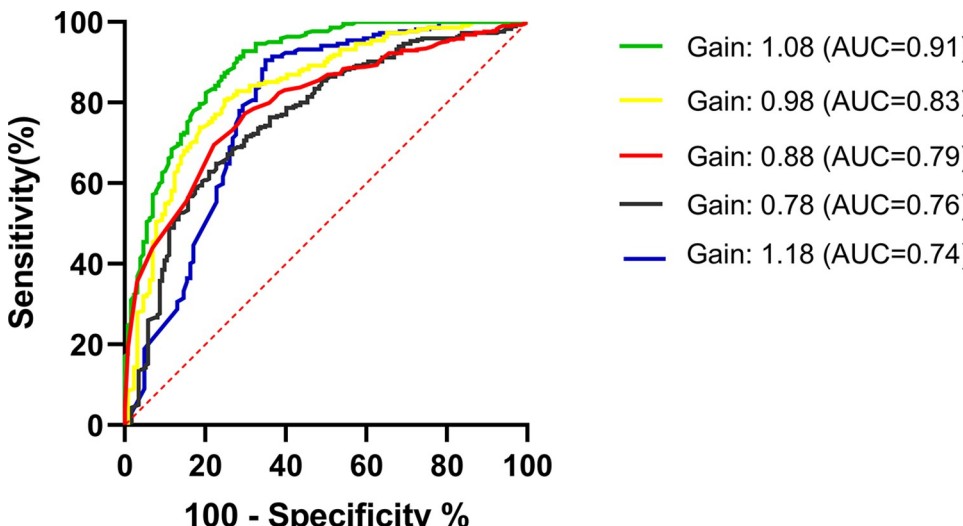

**Fig 4. The performance of urine flow cytometry in screening for urinary WBCs in pregnant women among different CR(WBC)-CW-FSC-P Gain values.**

**Table 4. The performance of urine flow cytometry among different CR(WBC)-CW-FSC-P Gain values in screening urinary WBC of pregnant women.**

|  | Gain values | | | | |
|---|---|---|---|---|---|
|  | **0.78** | **0.88** | **0.98** | **1.08** | **1.18** |
| **False positive rate (%)** | 35.13 | 30.43 | 18.91 | 9.45 | 6.75 |
| **False negative rate (%)** | 21.25 | 22.13 | 25.58 | 27.9 | 44.18 |
| **Youden index (%)** | 45.88 | 55.92 | 57.45 | 65.33 | 53.71 |

## Discussion

This study elucidated that the increased prevalence of UTIs among gravidae compared to non-pregnant females might be attributable to diagnostic errors. Subsequent investigations have indicated that these inaccuracies may originate from an elevated rate of false positives when utilizing UF to ascertain the presence of urinary WBCs in pregnant individuals. By calibrating the coefficient ratio (CR) of WBCs to the compensated forward scatter (CW-FSC-P) gain value within the UF protocol, the incidence of false positives during the detection of urinary WBCs in gravidae could be markedly diminished. Consequently, this adjustment could reduce the clinical misdiagnosis rate of UTIs among pregnant women. The implications of this discovery hold significant potential for clinical practice, particularly in enhancing the precision of UTI diagnoses within the obstetric population.

This study presents a pivotal finding concerning the elevated false-positive rate for UTIs in pregnant women, which was as high as 30.43% when employing UF. This heightened rate is likely due to the analyzer's tendency to erroneously classify ECs as WBCs, a phenomenon supported by the comparable false-negative rate for ECs and a notable linear correlation between the counts of these two cell types. López et al. [11] previously highlighted that UF exhibits a significant rate of false positives in the diagnosis of UTIs among pregnant women, aligning with the current study's observations. Notably, the research conducted by Kim et al. [12] also suggests that UF has a high false-positive rate in diagnosing UTIs within the general population. Incorrect diagnosis of UTIs carries significant implications for both individual patients and the healthcare system as a whole. The overdiagnosis of UTIs, particularly in cases of false positives, can lead to unnecessary antibiotic prescriptions. This overuse of antibiotics contributes to the growing problem of antibiotic resistance, where bacteria evolve to become less susceptible to the effects of antibiotics, making infections more difficult to treat [13, 14]. Such misdiagnosis may engender unwarranted anxiety among patients, particularly when they undergo additional, unrequired testing and therapeutic interventions under the erroneous presumption of a UTI. This scenario can engender discomfort and potential iatrogenic harm due to the imposition of superfluous medical procedures and pharmacological agents. Furthermore, misdiagnosis can precipitate augmented healthcare expenditures, attributable to the costs of superfluous diagnostics, therapeutics, and hospital stays, imposing a significant financial strain on both the patient and the healthcare infrastructure. The erosion of patient trust in healthcare providers and the medical establishment is another critical consequence of incorrect diagnoses. Broadly, the misdiagnosis of UTIs can impede public health initiatives focused on the containment of infectious diseases and the advocacy of judicious antibiotic stewardship. The precision of diagnostic practices is indispensable for the efficacious surveillance and management of UTIs within the population. It is imperative to address these challenges through rigorous clinical governance, enhanced diagnostic protocols, and the promotion of antibiotic resistance mitigation strategies. Consequently, there is an imperative need for a diagnostic approach that can diminish the false-positive rate associated with UF and enhance its diagnostic precision, thereby mitigating the misdiagnosis rate of UTIs in pregnant women. Bilsen

et al. [15] identified that the current pyuria thresholds are set too low, leading to an inappropriate diagnosis of UTIs in older women. By elevating these thresholds, they demonstrated a reduction in the misdiagnosis rate of UTIs among the elderly. However, the entrenched practice of utilizing current pyuria cutoffs for UTI diagnosis by physicians presents a challenge in altering this clinical approach. Therefore, the objective of this study was to augment the accuracy of UTI diagnosis in pregnant women without necessitating departure from the traditional pyuria cutoffs.

This study provides an in-depth analysis of the diagnostic markers for UTIs, with a particular focus on the utility of UF in different patient demographics. In the field of medical diagnostics, UF, as a relatively new technique, has been extensively applied in the clinical diagnosis of UTIs. Compared to conventional methods, UF offers distinct advantages, including high-throughput processing that facilitates the simultaneous analysis of multiple samples with rapid turnaround times. This technique is adept at swiftly identifying critical urinary components such as leukocytes, erythrocytes, and epithelial cells, thereby conferring a high degree of sensitivity and specificity in the diagnosis of UTIs. The automation inherent in UF significantly diminishes human operational errors, enhancing the uniformity and dependability of the test results. However, UF is not without its drawbacks. The substantial costs associated with the equipment and requisite reagents can be prohibitive, potentially precluding its adoption in settings with economic constraints. The traditional criterion of WBC count exceeding 5/HPF in urine remains a prevalent indicator for UTIs, despite the superior accuracy offered by urine bacterial culture, which is considered the gold standard. However, the requirement for a substantial time investment and the susceptibility to false negatives make the latter less suitable for rapid clinical diagnosis [16–18].

Our research findings, corroborated by the analysis of ROC curves, revealed that the diagnostic efficacy of UF is attenuated in pregnant women when compared to non-pregnant women. Nonetheless, the WBC count criterion of greater than 5/HPF in urine maintains substantial diagnostic relevance for UTIs in non-pregnant women. The heightened sensitivity of UF could be a contributing factor to the elevated false-positive rate [19, 20]. Modulating the sensitivity of UF presents a potential strategy to mitigate the false-positive rate of WBC detection, thereby reducing the misdiagnosis rate of UTIs in pregnant women, without the need to modify existing cutoffs. Considering the superior diagnostic performance of UF in non-pregnant women with UTIs and its diminished effectiveness in pregnant women, it is plausible that the analyzer's parameters necessitate fine-tuning for distinct patient populations. By adjusting the Gain value of CR(WBC)-CW-FSC-P, we have successfully achieved a marked reduction in the false-positive rate for WBC detection in pregnant women. This modification has led to an enhanced Youden's index, signifying an improved equilibrium between sensitivity and specificity, and consequently, a more precise diagnosis of UTIs within this demographic.

Our results indicate that a Gain value of 1.08 is optimal for minimizing the false-positive rate and maximizing the diagnostic efficacy of UF in the detection of UTIs in pregnant women. This targeted optimization offers a straightforward and efficacious solution to enhance the accuracy of UTI diagnosis during pregnancy, thereby potentially improving patient outcomes and reducing unnecessary antibiotic use.

## Limitations and future research

The sample size, though substantial, may not be fully representative of all pregnant women, and the study was conducted at a single medical center. Future research should aim to validate our findings in a larger population and across multiple centers. Additionally, the long-term clinical outcomes associated with the misdiagnosis of UTIs in pregnancy warrant further investigation.

## Conclusions

In conclusion, our study underscores the possibility that the incidence of UTIs among pregnant women is potentially overstated due to the inherent limitations of UF. Through the identification of the misclassification of ECs as WBCs as a contributing factor, and by proposing a refinement in the parameters of the UF analyzer, we have delineated a methodological approach to mitigate the occurrence of misdiagnosis. This adjustment not only enhances the diagnostic accuracy of UF but also has significant implications for optimizing clinical management and reducing unnecessary treatment in this patient population.

## Supporting information

**S1 Table. Correlations between age, gestational age, red blood cells, white blood cells, yeast, bacteria, epithelium.**
(DOCX)

## Author Contributions

**Conceptualization:** Zhaojie Liu, Dan Liu, Guangming Su, Wei Yang.

**Data curation:** Zhaojie Liu, Dan Liu, Guangming Su, Wei Yang.

**Formal analysis:** Dan Liu, Guangming Su, Wei Yang.

**Writing – original draft:** Zhaojie Liu, Wei Yang.

**Writing – review & editing:** Wei Yang.

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
