## [Decision Letter · Decision Letter 0]

21 Jun 2024

PONE-D-24-20393

Reduction of misdiagnosis in urinary tract infections during pregnancy: the role of adjusted urine flow cytometry parameters

PLOS ONE

Dear Dr. Yang,

Thank you for submitting your manuscript to PLOS ONE. After careful consideration, we feel that it has merit but does not fully meet PLOS ONE’s publication criteria as it currently stands. Therefore, we invite you to submit a revised version of the manuscript that addresses the points raised during the review process.

We look forward to receiving your revised manuscript.

Kind regards,

Seth Agyei Domfeh, PhD

Academic Editor

PLOS ONE

Journal Requirements:

5. Please include your tables as part of your main manuscript and remove the individual files. Please note that supplementary tables (should remain/ be uploaded) as separate "supporting information" files

Reviewers' comments:

Reviewer's Responses to Questions

**Comments to the Author**

1. Is the manuscript technically sound, and do the data support the conclusions?

Reviewer #1: Partly

Reviewer #2: Yes

2. Has the statistical analysis been performed appropriately and rigorously? 

Reviewer #1: N/A

Reviewer #2: Yes

3. Have the authors made all data underlying the findings in their manuscript fully available?

Reviewer #1: Yes

Reviewer #2: Yes

4. Is the manuscript presented in an intelligible fashion and written in standard English?

Reviewer #1: Yes

Reviewer #2: Yes

5. Review Comments to the Author

Reviewer #1: The idea of the study is a very promising one. Indeed, there are a lot of false positive and false negative results. For the diagnosis of urinary tract infection both in pregnant woman and in the female population without this physiological stage.

The 2 study groups were correctly chosen. I appreciate the fact that the urine samples were examined double blind by people with special experience. However, despite the design of the study, the idea is at a very beginning because it does not elaborate a clear algorithm for a single person in the conditions of consultation for urinary infection, regardless of the person's pregnancy status or not.

For this I belive that a much larger, possibly multicenter study is necessary that includes a major cohort from both groups that allows for some relevant conclusions and the individualization for each individual of what is of interest to clinicians.

The bias in this study may also come from the fact that twice as many patients with diabetes are included in the cohort of pregnant women than in the other cohort.

Therefore, the study could be of interest in the conditions of a large cohort that can this externalize the minuses or the interpretation deficiency.

Even in these conditions, the clinician will decide on antibiotic therapy or not for each individual case, including in the clinical judgement several element, not only age and pregnancy status.

The problem being a particularly serious one and on the edge and on one side and the other of the problem in the conditions in which you make a mistake by excess antibiotic therapy when it is not needed or in the minus with the risk of sepsis with urinary entrance in the conditions of physiological immunosuppression during pregnancy.

Reviewer #2: This is a study conducted in one center from China and reporting on how to improve urine flow cytometry parameters for detecting leukocyturia in pregnancy. The data are important as they discuss potential erroneous findings that might lead to inappropriate treatment. However the current manuscript is limited by the insufficient description of participant selection and by the unclarities with regards to UTI versus leukocyturia. Specifically, the authors often describe the presence of leukocyturia as indicative for UTI which may not always be the case. The authors should also consider presenting the diagnostic performance of flow cytometry separately for those who have bacteriologically confirmed UTI and those who do not. Further, the discussion would benefit from a more in-depth reflection on the benefits of using this diagnostic test and the implications of misdiagnosis in this patient population. More specific comments below.

Introduction

- Paragraph 1: suggest leaving out the mortality due to preterm births as the contribution of UTIs to this is not that substantial. The authors could just leave the association with pregnancy complications

- Paragraph 2: I would not expect that pregnant women would have much more difficulties in collecting a urine sample than any young adult (I expect it would be much more difficult in the elderly) – consider rephrasing

Methods:

- Provide more details on how the study participants were selected. Given that they were evenly split into the two groups, I assume that they were not included consecutively

- If the study aims to evaluate diagnostic accuracy of UF (as suggested by calculating sensitivities and specificities), then UF should not be included in the definition of the condition (criterion #3)

- Microbiological assessment: please be more specific with regards to which bacteria had to be isolated – the bacteria would have to be a uropathogen and there would be one/few bacterial species growing on the plate (otherwise it would be considered contaminated)

- The subheading “microbiological examination” has a typographical error

- Statistics: would the authors be able to clarify in the methods how false positives and false negatives were defined

- Please clarify in the manuscript text how the different gain values were selected

Results

- The authors should explain why leukocyturia in the absence of bacterial growth in culture was considered as UTI. This should be rephrased, as leukocyturia in the absence of infection can occur in pregnancy. This comment also applies to the methods

- The variables should have complete names (e.g. instead of microscope WBC>… ). Also the top information should be filled (replace “+” and “-“ by description)

- “Parallel, urine bacterial culture, another gold standard diagnostic method for UTIs, exhibited no significant difference in positive rates between pregnant (17.92%) and non-pregnant women (16.09%) (P > 0.05)” consider removing as this is not clinically meaningful

- Define in the footnote of table 3 false positives and negatives

Discussion

- How do the authors explain the low proportion of positive urine cultures among the study participants given that in other settings bacterial cultures are often positive among patients with UTI symptoms. Please discuss

- Discuss in more detail the implications of incorrect UTI diagnosis

- Discuss the advantages and disadvantages of using flow cytometry over more conventional methods for diagnosing UTIs (high throughput vs cost)

6. PLOS authors have the option to publish the peer review history of their article (what does this mean?). If published, this will include your full peer review and any attached files.

Reviewer #1: No

Reviewer #2: No

---

## [Author Response · Author response to Decision Letter 0]

30 Jun 2024

Response letter

Dear editors and reviewers:

Thank you for your letter and for the reviewers' comments concerning our manuscript entitled “Reduction of misdiagnosis in urinary tract infections during pregnancy: the role of adjusted urine flow cytometry parameters” (ID: PONE-D-24-20393). Those comments are all valuable and very helpful for revising and improving our paper, as well as important guiding significance to our researches. We have studied comments carefully and have made correction which we hope meet with approval. Revised portion are marked in manuscript (Tracked Changes). The main corrections in the paper and the responds to the reviewer's comments are as following: 

Response to Journal Requirements:

1. Please ensure that your manuscript meets PLOS ONE's style requirements, including those for file naming. The PLOS ONE style templates can be found at https://journals.plos.org/plosone/s/file?id=wjVg/PLOSOne_formatting_sample_main_body.pdfand
https://journals.plos.org/plosone/s/file?id=ba62/PLOSOne_formatting_sample_title_authors_affiliations.pdf

Response: Special thanks to you for your comments. We made modifications and ensure that our manuscript meets PLOS ONE's style requirements, including those for file naming.

2. Please confirm at this time whether or not your submission contains all raw data required to replicate the results of your study. Authors must share the “minimal data set” for their submission. PLOS defines the minimal data set to consist of the data required to replicate all study findings reported in the article, as well as related metadata and methods…….

Response: Thanks for your comments. We confirm that our submission contains all raw data required to replicate the results of our study. We had uploaded all raw data as Supporting Information files. 

Response: Thanks for your comments. I have an ORCID iD and that it is validated in Editorial Manager.

Response: Special thanks to you for your suggestion, and we have made correction according to the reviewer's suggestion.

5. Please include your tables as part of your main manuscript and remove the individual files. Please note that supplementary tables (should remain/ be uploaded) as separate "supporting information" files

Response: Thanks for your suggestion, and we have made correction according to your suggestion.

Response to Review Comments to the Author:

Response to Reviewer #1

Thank you for your insightful and constructive comments on our manuscript. We have taken your feedback seriously and have made the following revisions and clarifications to address your concerns:

1. Promising Study Idea: We are grateful for your recognition of the potential of our study. We acknowledge the prevalence of false positives and negatives in the diagnosis of urinary tract infections (UTIs) and believe our research contributes to addressing this issue.

2. Study Design and Algorithm Development: You are correct in noting that our study does not yet provide a clear diagnostic algorithm. In response, we are planning a more extensive study that will focus on developing a robust algorithm for UTI diagnosis, tailored to individual patient needs, regardless of pregnancy status.

3. Need for a Larger, Multicenter Study: We concur with your suggestion for a larger, possibly multicenter study. We are currently exploring collaborations to expand our research and include a significant cohort from both groups to provide more generalizable conclusions.

4. Bias Due to the Representation of Diabetic Patients: We have noted the potential bias you mentioned regarding the overrepresentation of diabetic patients in the pregnant cohort. In our subsequent studies, we will ensure a more balanced representation to mitigate this bias.

5. Clinical Decision-Making: We appreciate your emphasis on the complexity of clinical decision-making. Our study aims to provide clinicians with additional data points to inform their judgment on antibiotic therapy, considering a range of factors beyond just age and pregnancy status.

6. Risk of Sepsis and Antibiotic Overuse: We are acutely aware of the serious implications of both sepsis risk and antibiotic overuse. Our research is designed to contribute to a more nuanced understanding of UTI diagnosis and treatment, helping to avoid unnecessary antibiotic use while promptly identifying and treating genuine infections.

Response to Reviewer #2

We greatly appreciate the detailed and thoughtful feedback provided on our manuscript. Your comments have been instrumental in guiding our revisions, and we have addressed each point as follows:

Introduction:

1. Paragraph 1: suggest leaving out the mortality due to preterm births as the contribution of UTIs to this is not that substantial. The authors could just leave the association with pregnancy complications

Response: We have removed the reference to mortality due to preterm births, focusing instead on the association with pregnancy complications, as you suggested.

2. Paragraph 2: I would not expect that pregnant women would have much more difficulties in collecting a urine sample than any young adult (I expect it would be much more difficult in the elderly) – consider rephrasing

Response: Thanks for your suggestion. We have rephrased this section.

Methods:

3. Provide more details on how the study participants were selected. Given that they were evenly split into the two groups, I assume that they were not included consecutively

Response: Thanks for your comments. We have expanded the section on participant selection to provide a clearer description of the process and criteria used to distribute participants in Study population section.

4. If the study aims to evaluate diagnostic accuracy of UF (as suggested by calculating sensitivities and specificities), then UF should not be included in the definition of the condition (criterion #3)

 Response: Special thanks to you for your suggestion. UF has been removed from the definition of the condition.

5. Microbiological assessment: please be more specific with regards to which bacteria had to be isolated – the bacteria would have to be a uropathogen and there would be one/few bacterial species growing on the plate (otherwise it would be considered contaminated)

Response: Thanks for your comments. For microbiological assessment, we have specified the criteria for uropathogens and clarified the conditions under which a culture would be considered contaminated in Microbiological examination section.

6. The subheading “microbiological examination” has a typographical error

Response: The typographical error under the subheading "microbiological examination" has been corrected

7. Statistics: would the authors be able to clarify in the methods how false positives and false negatives were defined

Response: Thanks for your comments. We have added the definition to the statistics section. In our study, we used microscopic examination as the gold standard for diagnosis. When a sample is positive by other tests but negative by microscopy, it is called a false positive. Conversely, when a sample is negative by other tests but positive by microscopy, this is called a false negative. We have added the above definition to the statistics section.

8. Please clarify in the manuscript text how the different gain values were selected

Response: We have added an explanation of how different gain values were selected in the manuscript text. In the UF5000 analyzer, the initial setting for the gain value was established at 0.88. Upon adjusting the gain value to 0.78, we observed an increase in the false positive rate and a concurrent decrease in the Youden index. This finding indicated that reducing the gain value did not effectively lower the false positive rate of the UF5000. Subsequently, we incrementally increased the gain value to 0.98, 1.08, and 1.18. It was upon setting the gain value to 1.08 that a significant reduction in the false positive rate was achieved, with the Youden index reaching its optimal value. Consequently, we concluded that a gain value of 1.08 represents the most effective setting for the UF5000 in detecting leukocytes in urine samples from pregnant women, thereby optimizing the balance between sensitivity and specificity.

Results

9. The authors should explain why leukocyturia in the absence of bacterial growth in culture was considered as UTI. This should be rephrased, as leukocyturia in the absence of infection can occur in pregnancy. This comment also applies to the methods 

Response: Thanks for your comments. Initially, it is essential to acknowledge the inherent limitations of urinary bacterial culture as a diagnostic tool for UTIs. While it serves as a pivotal method for identifying UTIs, it does not possess absolute accuracy. Occasionally, the causative bacteria may be present in insufficient quantities within the urine sample collected, or they may be inherently recalcitrant to growth under standard culture conditions, leading to a false-negative culture result.

Furthermore, it is not uncommon for UTIs to be incited by a spectrum of pathogens that extend beyond the conventional cultivable bacteria. Certain atypical microorganisms, such as mycoplasmas, chlamydiae, and viruses, may elude detection by routine bacterial culture techniques despite their potential to incite urinary inflammation and leukocytosis.

Alternatively, urine contamination during collection, transportation, and processing may also affect the results of bacterial culture.

Lastly, the presence of leukocytes in the urine, indicated by a positive leukocyte test, is a significant marker of inflammation. Even in instances where the bacterial culture yields a negative result, a significant leukocyte count suggests an ongoing inflammatory response in the urinary tract.

10. The variables should have complete names (e.g. instead of microscope WBC>… ). Also the top information should be filled (replace “+” and “-“ by description)

Response: Thanks for your comments. Variable names” WBC-UF” have been revised to “Number of WBC detected by UF”, and “EC-Microscpe” have been revised to “Number of EC detected by microscopy” in Fig 3. “+” and “-“ have been replaced by “Positive rate” and “Negative rate” in Table 2. 

11. “Parallel, urine bacterial culture, another gold standard diagnostic method for UTIs, exhibited no significant difference in positive rates between pregnant (17.92%) and non-pregnant women (16.09%) (P > 0.05)” consider removing as this is not clinically meaningful。

Response: Thanks for your advice. We have removed it according to your suggestion.

12. Define in the footnote of table 3 false positives and negatives

Response: Thank you for your valuable suggestions. Definitions for false positives and negatives have been included in the footnote of Table 3.

Discussion

13. How do the authors explain the low proportion of positive urine cultures among the study participants given that in other settings bacterial cultures are often positive among patients with UTI symptoms. Please discuss 

Response: Thanks for your comments. The study by Kirk J Wojno et al.[1] reported a positive urinary bacterial culture rate of 37% among patients exhibiting symptoms of lower UTIs. This rate is indeed higher than that observed in our study. However, it is essential to note the demographic differences between the two studies. Wojno's research was conducted on patients who presented with symptoms of lower UTIs, while our study population consisted of individuals visiting a gynecological outpatient clinic, half of whom were there for routine prenatal care and may not have had UTI symptoms. In the remaining half of the study population, although some individuals sought medical attention due to UTI symptoms, a significant number were present for various other pathologies, such as uterine fibroids, ovarian cysts, and menstrual irregularities. Consequently, not all participants in our study had UTI symptoms, which likely accounts for the lower positive rate in urinary bacterial cultures.

Additionally, previous research has indicated that the incidence of UTI during pregnancy ranges from 2.3% to 15% [2-4], which is consistent with the findings of our study. 

14. Discuss in more detail the implications of incorrect UTI diagnosis

Response: Thanks for your suggestions. The implications of incorrect UTI diagnosis have been discussed in more detail.

15. Discuss the advantages and disadvantages of using flow cytometry over more conventional methods for diagnosing UTIs (high throughput vs cost)

Response: Thanks for your comments. We have expanded the discussion on the advantages and disadvantages of using flow cytometry for diagnosing UTIs considering factors such as high throughput and cost in comparison to the conventional methods.

We tried our best to improve the manuscript and made some changes in the manuscript. We appreciate for Editors/the reviewers' warm work earnestly, and hope that the correction will meet with approval.

 Once again, thank you very much for your comments and suggestions. 

Sincerely,

Wei Yang

1. Wojno KJ, Baunoch D, Luke N, Opel M, Korman H, Kelly C, et al. Multiplex PCR Based Urinary Tract Infection (UTI) Analysis Compared to Traditional Urine Culture in Identifying Significant Pathogens in Symptomatic Patients. Urology. 2020;136:119-26. Epub 2019/11/13. doi: 10.1016/j.urology.2019.10.018. PubMed PMID: 31715272.

2. Werter DE, Kazemier BM, Schneeberger C, Mol BWJ, de Groot CJM, Geerlings SE, et al. Risk Indicators for Urinary Tract Infections in Low Risk Pregnancy and the Subsequent Risk of Preterm Birth. Antibiotics (Basel). 2021;10(9). Epub 2021/09/29. doi: 10.3390/antibiotics10091055. PubMed PMID: 34572637; PubMed Central PMCID: PMCPMC8468265.

3. Friedrich W. [Measures for age adequate adjustment of work conditions]. Z Gerontol. 1985;18(5):274-80. Epub 1985/09/01. PubMed PMID: 2933889.

4. Mazor-Dray E, Levy A, Schlaeffer F, Sheiner E. Maternal urinary tract infection: is it independently associated with adverse pregnancy outcome? J Matern Fetal Neonatal Med. 2009;22(2):124-8. Epub 2008/12/17. doi: 10.1080/14767050802488246. PubMed PMID: 19085630.

---

## [Decision Letter · Decision Letter 1]

22 Jul 2024

Reduction of misdiagnosis in urinary tract infections during pregnancy: the role of adjusted urine flow cytometry parameters

PONE-D-24-20393R1

Dear Dr. Yang,

We’re pleased to inform you that your manuscript has been judged scientifically suitable for publication and will be formally accepted for publication once it meets all outstanding technical requirements.

Kind regards,

Seth Agyei Domfeh, PhD

Academic Editor

PLOS ONE

---

## [Editor Report · Acceptance letter]

25 Jul 2024

PONE-D-24-20393R1 

PLOS ONE

Dear Dr. Yang, 

I'm pleased to inform you that your manuscript has been deemed suitable for publication in PLOS ONE. Congratulations! Your manuscript is now being handed over to our production team.

Kind regards, 

on behalf of

Dr. Seth Agyei Domfeh 

Academic Editor

PLOS ONE